# Inflating hollow nanocrystals through a repeated Kirkendall cavitation process

He Tianou[1,4], Weicong Wang[1], Xiaolong Yang[1,2], Zhenming Cao[3], Qin Kuang[3], Zhao Wang[1,2], Zhiwei Shan [4], Mingshang Jin [1] & Yadong Yin [5]

The Kirkendall effect has been recently used to produce hollow nanostructures by taking advantage of the different diffusion rates of species involved in the chemical transformations of nanoscale objects. Here we demonstrate a nanoscale Kirkendall cavitation process that can transform solid palladium nanocrystals into hollow palladium nanocrystals through insertion and extraction of phosphorus. The key to success in producing monometallic hollow nanocrystals is the effective extraction of phosphorus through an oxidation reaction, which promotes the outward diffusion of phosphorus from the compound nanocrystals of palladium phosphide and consequently the inward diffusion of vacancies and their coalescence into larger voids. We further demonstrate that this Kirkendall cavitation process can be repeated a number of times to gradually inflate the hollow metal nanocrystals, producing nanoshells of increased diameters and decreased thicknesses. The resulting thin palladium nanoshells exhibit enhanced catalytic activity and high durability toward formic acid oxidation.

[1] Frontier Institute of Science and Technology and State Key Laboratory of Multiphase Flow in Power Engineering, Xi'an Jiaotong University, Xi'an, Shaanxi 710049, China. [2] Department of Physics, Guangxi Key Laboratory for Relativistic Astrophysics, Guangxi University, Nanning 530004, China. [3] Department of Chemistry, State Key Laboratory of Physical Chemistry of Solid Surfaces, College of Chemistry and Chemical Engineering, Xiamen University, Xiamen 361005, China. [4] Center for Advancing Materials Performance from the Nanoscale, State Key Laboratory for Mechanical Behavior of Materials, Xi'an Jiaotong University, Xi'an 710049, China. [5] Department of Chemistry, University of California, Riverside, CL 92521, USA. Tianou He, Weicong Wang and Xiaolong Yang contributed equally to this work. Correspondence and requests for materials should be addressed to M.J. (email: jinm@mail.xjtu.edu.cn) or to Y.Y. (email: yadong.yin@ucr.edu)

The Kirkendall effect, a classical metallurgy phenomenon, has recently emerged as a unique self-templated strategy for the preparation of hollow nanocrystals with high yield and good control over size and shape[1–8]. Since we first employed this mechanism to explain the formation of hollow nanocrystals of cobalt oxides and chalcogenides[1], extensive efforts have been devoted to popularizing the Kirkendall effect in fabricating hollow nanostructures of diverse morphologies (i.e., spheres[9–11], nanotubes[12–14], and dendrites[15, 16]) and compositions (i.e., bimetallic alloys[17], non-metallic oxides[18], and ternary systems[19]). Typically, Kirkendall voids form during the reaction of solid nanocrystals with species in their surroundings, where the preferred diffusion of atoms or ions from core material to the reaction interface leads to an unbalanced outward material flux and simultaneously the production and coalescence of vacancies inside the nanocrystals.

In relatively simpler cases involving reaction of metal (M) nanocrystals with a species X in the surrounding, a shell of compound MX is formed at the reaction interface. When the outward diffusion of M is faster than the inward diffusion of X, Kirkendall voids form within the nanocrystals of MX[11]. In contrast, in the case where the inward diffusion of X is faster, the final product of MX remain as solid nanocrystals. An interesting prediction in the latter case is that if X can be extracted effectively from the MX nanocrystals, the outward diffusion of X may be still faster than the inward diffusion of M, so vacancies will be injected into the core, again resulting in Kirkendall voids. The major advantage of this reverse process is that the Kirkendall voids are produced inside monometallic nanocrystals rather than compound nanocrystals as demonstrated in the previous cases. Such a possibility is fascinating in the field of colloidal nanostructure synthesis as hollow nanocrystals of metals have shown great advantages in various applications, thanks to their unique catalytic, optical, and magnetic properties[20–29].

In this work, we demonstrate that Kirkendall voids can be formed by extracting a species X from solid nanocrystals of a compound MX, allowing the production of monometallic hollow nanocrystals. Solid metal nanocrystals of M, which become solid nanocrystals of MX after reacting with species X due to the faster diffusion of X than M, can be eventually converted to hollow metal nanocrystals with an increased outer diameter through an artful application of the Kirkendall effect by extracting the species

X. We further demonstrate that this Kirkendall cavitation process can be repeated a number of times to gradually inflate the hollow metal nanocrystals, producing shells of increased diameters and decreased thicknesses. We attribute the key to the success of this process in producing monometallic hollow nanocrystals to the effective extraction reaction that can promote the outward diffusion of species X from the compound nanocrystals of MX. The resulting thin M nanoshells may exhibit enhanced catalytic activity and high durability toward catalytic reactions.

## Results

**Synthesis and characterization of hollow palladium nanocrystals.** The primary process and the underlying concept are schematically illustrated in Fig. 1. Here we chose palladium (Pd) and its phosphide as a proof of concept system by considering the relatively weak binding between Pd atoms[30], which should benefit the insertion of other elements and the well-studied phosphorization reaction of Pd[31–34]. Experimentally, we started with the synthesis of uniform Pd nanocubes using a method that we developed previously[35–37], and then use these nanocubes to react with trioctylphosphine (TOP) at 250 °C to produce phosphide nanocrystals. Bright-field transmission electron microscopy (TEM) images in Fig. 2a, b show that the original Pd nanocubes with an average edge length of 18 nm were transformed into spherical solid phosphide particles with an average diameter of ~ 20 nm. Volume wise, the phosphide particles are slightly smaller than the initial Pd nanocubes, which could be attributed to minor etching by TOP[38, 39]. High-resolution TEM (HRTEM) imaging and X-ray diffraction (XRD) measurements (Supplementary Fig. 1) indicate that Pd nanocrystals have turned into an amorphous phase after reacting with TOP, confirming the disruption to the crystal lattice when phosphorus (P) atoms are inserted into the original Pd nanocubes. This significant lattice disruption also contributes to the shape transformation from cubes to spheres during the phosphorization. The atomic molar ratio of P:Pd determined by inductively coupled plasma mass spectrometry (ICP-MS) is 1.98:1 for the amorphous particles (Supplementary Table 1), suggesting their composition is close to $PdP_2$, which is consistent with previous works[34]. The energy-dispersive X-ray spectroscopy (EDS) elemental mapping images and line-scanning profile of the $PdP_2$ intermediates demonstrate

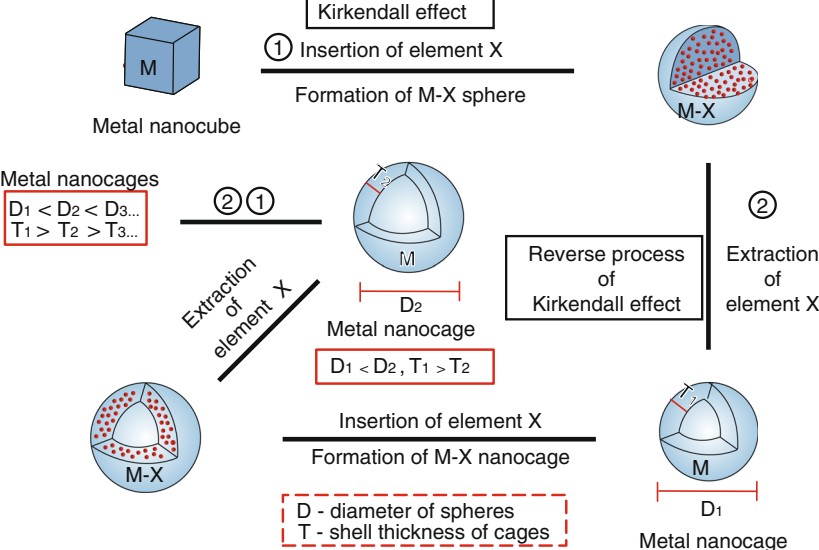

**Fig. 1** Schematic illustration of the synthetic strategy of hollow metal nanocrystals. Schematic illustration of the repeated Kirkendall cavitation process that leads to the formation and enlargement of hollow monometallic nanocrystals

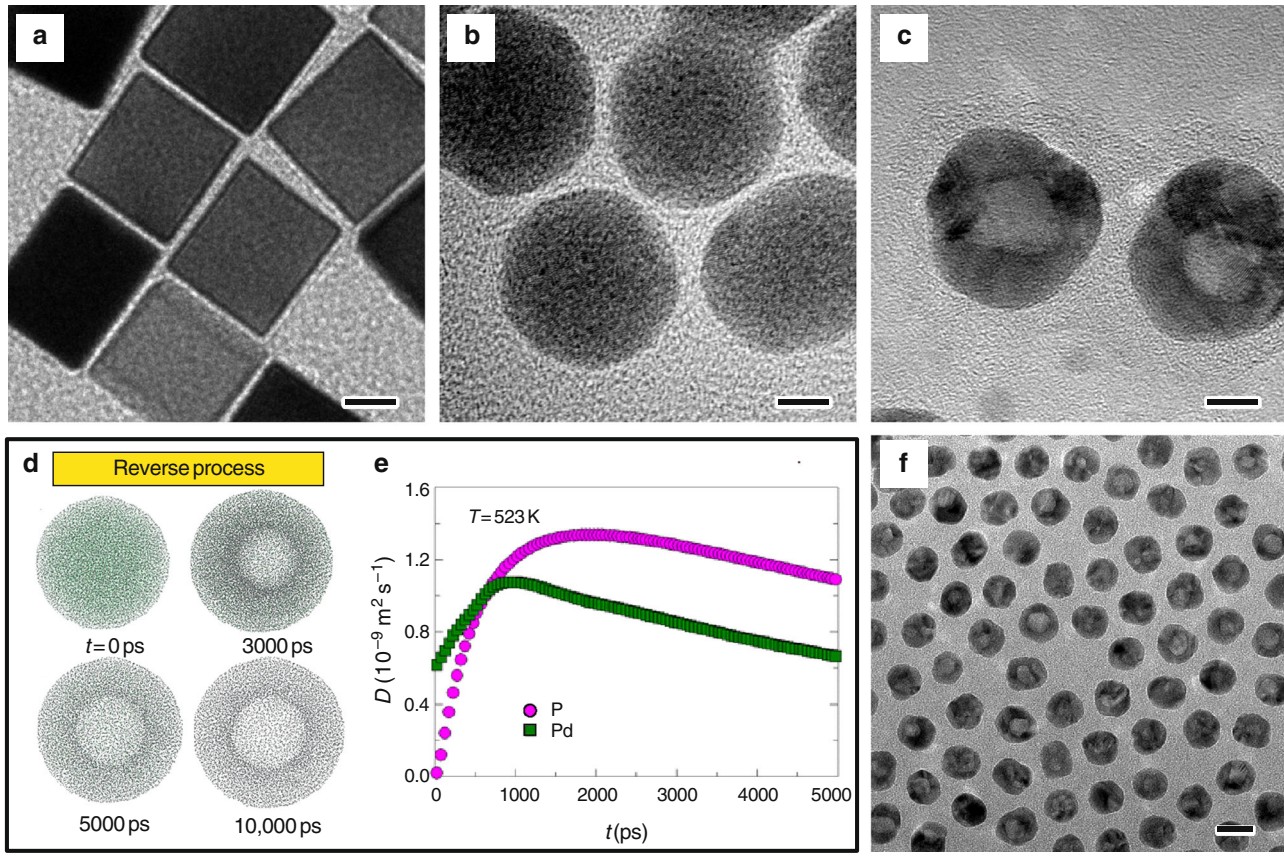

**Fig. 2** Transformation of solid Pd nanocubes into Pd hollow nanocrystals with hollow interiors. **a–c** TEM images of Pd nanocubes, Pd-P intermediates, and hollow Pd nanospheres, respectively. Scale bars in **a–c** are 5 nm. **d** Snapshots of a Pd nanocrystal at different simulation time. **e** Instantaneous diffusion coefficients $D$ of Pd and P atoms during the simulation. **f** TEM images of hollow Pd nanospheres at a low magnification. Scale bar in **f** is 20 nm

that both Pd and P are distributed homogeneously within the particles (Supplementary Figs. 2 and 3). These PdP₂ amorphous nanoparticles then serve as intermediates for the following extraction reaction to generate hollow Pd nanocrystals.

Producing hollow Pd nanocrystals requires an extraction reaction to drive the outward diffusion of P atoms, which can be achieved by reacting the nanocrystals with oxygen at elevated temperatures. Experimentally, the amorphous PdP₂ nanoparticles were cleaned, re-dispersed in oleylamine (OAm), and then heated at 250 °C under air. Interestingly, hollow particles with an average pore size of 7 nm were generated, as demonstrated in Fig. 2c. HRTEM image and XRD pattern (Supplementary Fig. 4) suggest that the hollow particles were composed of polycrystalline domains. The molar ratio of P:Pd decreased to 1:8.8 in the resulting hollow Pd nanocrystals (Supplementary Table 1). The small amount of P in the system may be partly attributed to the residual TOP molecules adsorbed on the Pd surface[40], as the molar ratio of P:Pd was found to further decrease to 1:11.8 if the nanoparticles were washed with ethanol twice. This understanding was further confirmed by Fourier transform infrared spectroscopy analysis of the washed nanoparticles, which revealed the existence of TOP molecules (Supplementary Fig. 5). Compared with initial Pd nanocubes and Pd-P intermediates, the average size of hollow Pd nanocrystals increased to 22 nm, showing an increase in an overall dimension similar to previously reported Kirkendall cavitation processes[1]. According to previous studies, PdP₂ nanoparticles are stable at the reaction temperature, ruling out the possible decomposition route to the metallic phase[41]. On the other hand, PdP₂ can react with O₂, giving the formation of P₂O₅ under thermal treatment[42]. The resulting P₂O₅

is believed to react with OAm, which is the solvent and in large excess. It is believed that the rapid oxidation of P at the high reaction temperature provided an effective driving force for the dominant outward diffusion of P. A control experiment was carried out by heating the cleaned PdP₂ nanoparticles in OAm at 250 °C under nitrogen instead of air. Spherical solid polycrystal-line Pd nanocrystals with an average diameter of 18.5 nm were obtained, as shown in Supplementary Fig. 6. ICP-MS measurements suggested a P:Pd molar ratio of 1:8, again suggesting the extraction of P from the PdP₂ nanoparticles during the heating process. These results confirm the critical role of O₂ in promoting the outward diffusion of P. When the system was heated in air, the concentration of O₂ was relatively high so that the oxidation reaction was fast enough to realize the condition of $v_P > v_{Pd}$, which was required to generate hollow structures. In contrast, when the reaction was performed under nitrogen, the concentration of O₂ in the system was significantly lower as only a limited amount of O₂ originally dissolved in the reaction solution was available for the oxidation reaction. As a result, the required condition of $v_{P} > v_{Pd}$ could not be established so that the extraction reaction only led to solid Pd particles. This synthetic strategy was also effective for Pd nanocrystal precursors of other shapes. For example, when decahedral Pd nanocrystals were used as starting materials, hollow Pd nanocrystals can still be obtained after the reaction (Supplementary Fig. 7).

To attain further insight into the formation mechanism of the hollow structures, we performed atomistic simulations using the classical parallel molecular dynamics package LAMMPS[43]. The inter-atomic interactions were described by the embedded-atom-method potential with parameters fitted from the total

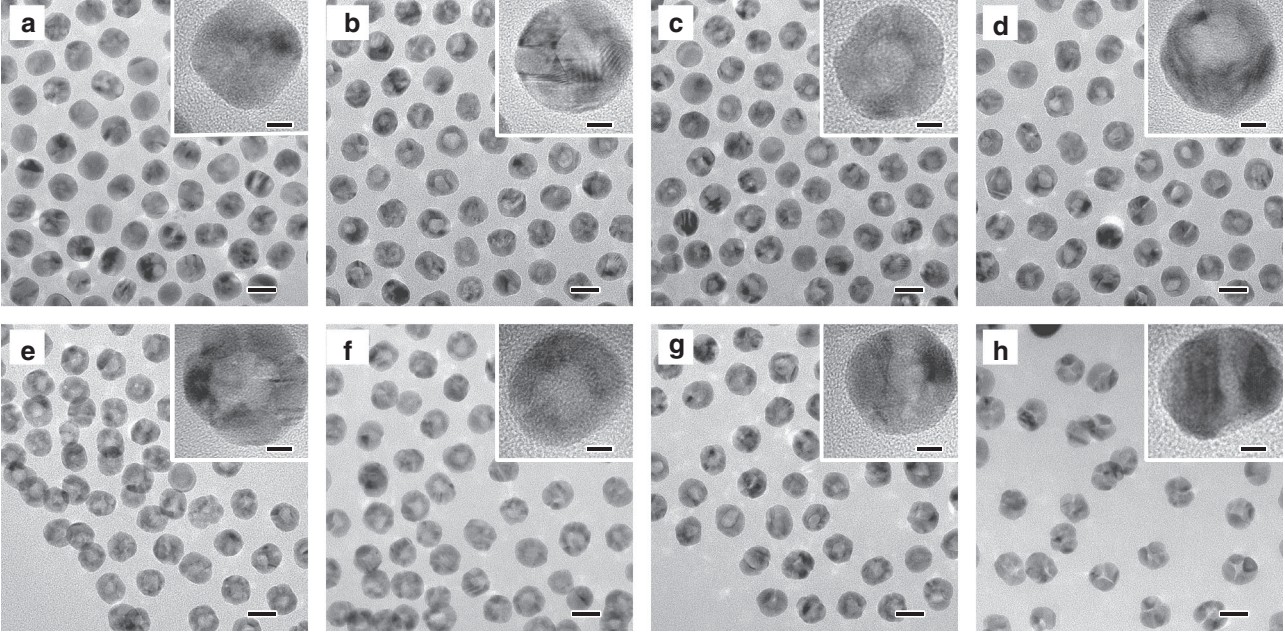

**Fig. 3** TEM and HRTEM images of hollow Pd hollow nanocrystals obtained with varied molar ratios of $N_2$:$O_2$: **a** 90:10, **b** 85:15, **c** 75:25, **d** 70:30, **e** 60:40, **f** 40:60, **g** 20:80, **h** 0:100. Scale bars are 20 nm. Scale bars in the insets are 5 nm

energy of a face-centered cubic unit cell as a function of the lattice constant (See Supplementary Methods, Supplementary Table 2 and Supplementary Fig. 8). We constructed a $PdP_2$ nanoparticle in three-dimensional space according to the experimental stoichiometry (Fig. 2d). To simulate the reaction between P and $O_2$ on the Pd-P surface, we artificially and progressively removed P atoms from the outer layer. The instantaneous diffusion coefficients ($D$) of Pd and P atoms were computed during the simulation. As shown in Fig. 2e, $D_P$ was higher than $D_{Pd}$ after deleting the P atoms for 600 ps (Supplementary Fig. 9). As a result, hollow structures formed, with the pore size increasing with time until P was completely removed (Fig. 2d), which was consistent with our experimental findings. Another notable fact was that $D_P$ was lower than $D_{Pd}$ at the initial stage before the outer layer P atoms reacted with $O_2$. As such, the diffusion of P was slower than Pd without the extra driving force by reacting with $O_2$ of sufficiently high concentration, which was in accordance with the fact that solid Pd nanocrystals rather than hollow ones were generated under the atmosphere of low $O_2$ concentration.

According to the theoretical study, the diffusivity evolution of the diffusion couple is crucial for the structure of final products. From this point of view, the reaction temperature is another important factor worth discussing as it determines the diffusion coefficient to a large extent[44, 45]. As shown in Supplementary Fig. 10, the final products appeared to hold more than one pores at lower reaction temperatures. In particular, porous Pd nanocrystals were generated when the temperature was lowered to 150 °C. This is consistent with the classical Kirkendall effect, in which supersaturated vacancies coalesce into many small voids at early stages and then progressively much larger ones, eventually to hollow particles if the system is a nanoparticle[1,2]. A considerably high temperature is therefore required in order to produce a sufficiently high concentration of vacancies and promote their significant condensation into large voids. At low temperatures, diffusion is slow so that much longer time is required for the small voids to coalesce, and they appear to be frozen at the small dimensions when observed at ordinary time scale.

To fully understand the impact of $O_2$ on the hollowing process, we performed a set of experiments with a varying volume ratio of $N_2$:$O_2$. As demonstrated in Fig. 3, the pore size increased with increasing $O_2$ concentration from 10 to 40%, owing to the enhanced oxidation rate of P induced by a higher concentration of $O_2$ and thereby a larger rate difference in the diffusion couple. The corresponding particles size and pore size distributions were shown in Supplementary Fig. 11. Noteworthy, the pore began to shrink when the $O_2$ content was further increased. Particularly, when pure $O_2$ was introduced, only narrow cracks were found to form in the Pd nanocrystals. This is understandable, considering that the fast reaction between P and $O_2$ may promote the outward diffusion of P and eventually lead to fracture of the particles. The rupture of the structure leads to the formation of channels through which $O_2$ molecules can enter the inner core directly, making it less likely to produce the directional mass diffusion that is required for cavitation through the Kirkendall effect. The ratios of Pd:P in the final products are measured to be around 7.0:1–8.8:1 (Supplementary Table 3).

**Synthesis and characterization of H-Pd-2 and H-Pd-3**. In principle, the Kirkendall cavitation process could be repeated to inflate the hollow nanocrystals into units with larger dimensions and thinner shells if there is an appropriate reaction to recover the shell to its original composition. In the previously reported cases involving the conversion of monometallic nanocrystals of M to hollow nanocrystals of compound MX, repeating the hollowing process has been challenging due to the lack of chemical reactions that can reduce MX back to M while maintaining the morphology of the hollow shells. The extraction reaction in the current work is therefore significant as it demonstrates the complete cycle of chemical conversion from Pd to $PdP_2$ and then back to Pd, accompanied by a morphological change from solid nanocrystals to hollow ones. Conceivably, one could repeat the cycle on the hollow nanocrystals and inflate these nanoscale "balloons" by further insertion and extraction of P to or from the shells, as schematically illustrated in Fig. 4a. Motivated by this new possibility, we repeated the cavitation process by starting with the

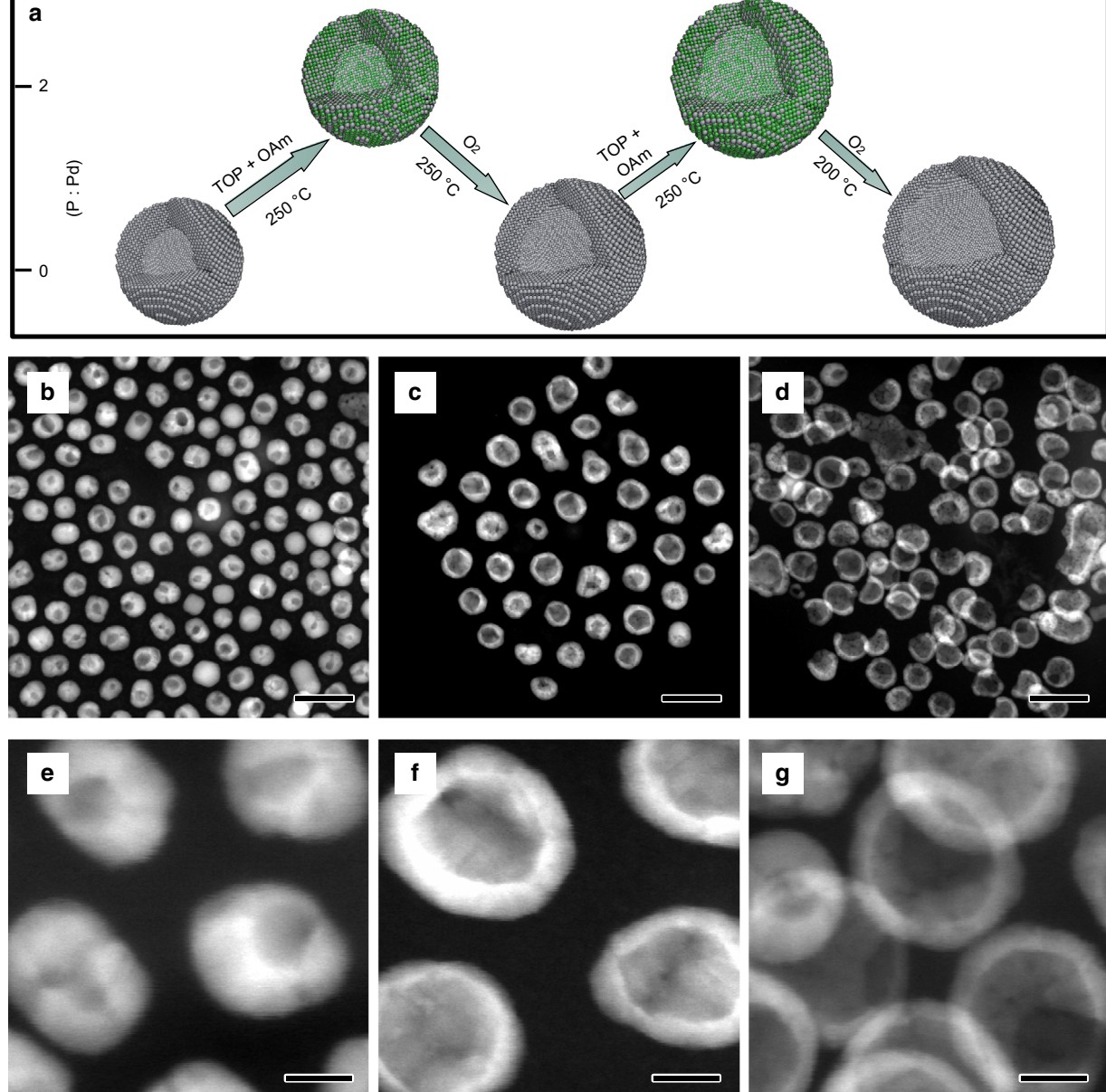

**Fig. 4** Synthesis of hollow Pd nanocrystals with thin walls by repeating the Kirkendall cavitation process three times. **a** Schematic illustration showing the evolution of the molar ratio of P:Pd during repeated cavitation cycles. **b–g** HAADF-STEM images, high-resolution HAADF-STEM images of the obtained hollow nanocrystals after repeated cavitation cycles: **b**, **e** H-Pd-1 obtained by one cavitation cycle; **c**, **f** H-Pd-2 achieved by two cavitation cycles; and **d**, **g** H-Pd-3 produced by three cavitation cycles. Scale bars in **b–d** are 50 nm. Scale bars in **e–g** are 5 nm

hollow Pd nanocrystals obtained after the first cycle (H-Pd-1, Fig. 4b, e, with an outer diameter of ~20.7 nm and pore size of ~7.5 nm). As expected, the polycrystalline hollow Pd nanocrystals were converted to amorphous hollow $PdP_2$ nanoshells after reacting with TOP (Supplementary Fig. 12), confirming the insertion of P to H-Pd-1 through the phosphorization reaction. It is important to note that the phosphorization reaction maintained the hollow shell morphology, implying that it was not a simple reverse process of the prior dephosphorization reaction. After the following step of removing P from the hollow $PdP_2$ nanoparticles, as shown in Fig. 4c, f, we were able to obtain hollow Pd nanocrystals (H-Pd-2) again. The insertion and extraction of P atoms during the second cycle were verified by ICP-MS measurements. The average outer diameter and pore size increased from ~20.7 nm and ~7.5 nm in the case of H-Pd-1 to 25.3 nm and 16 nm of H-Pd-2, respectively. More importantly,

the shell of the hollow nanocrystals became much thinner after the second cycle of the Kirkendall cavitation process. To further test the repeatability of the process, we carried out the insertion and extraction reactions again using the H-Pd-2 sample as the starting materials and produced Pd hollow nanocrystals with further increased outer diameter (~26.4 nm) and pore size (~19.2 nm), as demonstrated in Fig. 4d, g, and Supplementary Fig. 13. The shells of the hollow nanocrystals became as thin as 2–3 nm. Clearly, the shell thickness decreased as the diameter of hollow Pd nanocrystals increased, since no additional Pd source was introduced into the reaction system (Fig. 5). We further calculated the volume of the initial nanocubes and the obtained thin shells. Initially, the volume of a single nanocube is around 5800 $nm^3$. The volumes of Pd shells was calculated to be 4500 $nm^3$ for H-Pd-1, 5700 $nm^3$ for H-Pd-2, and 5800 $nm^3$ for H-Pd-3. These results indicated that Pd nanocubes were

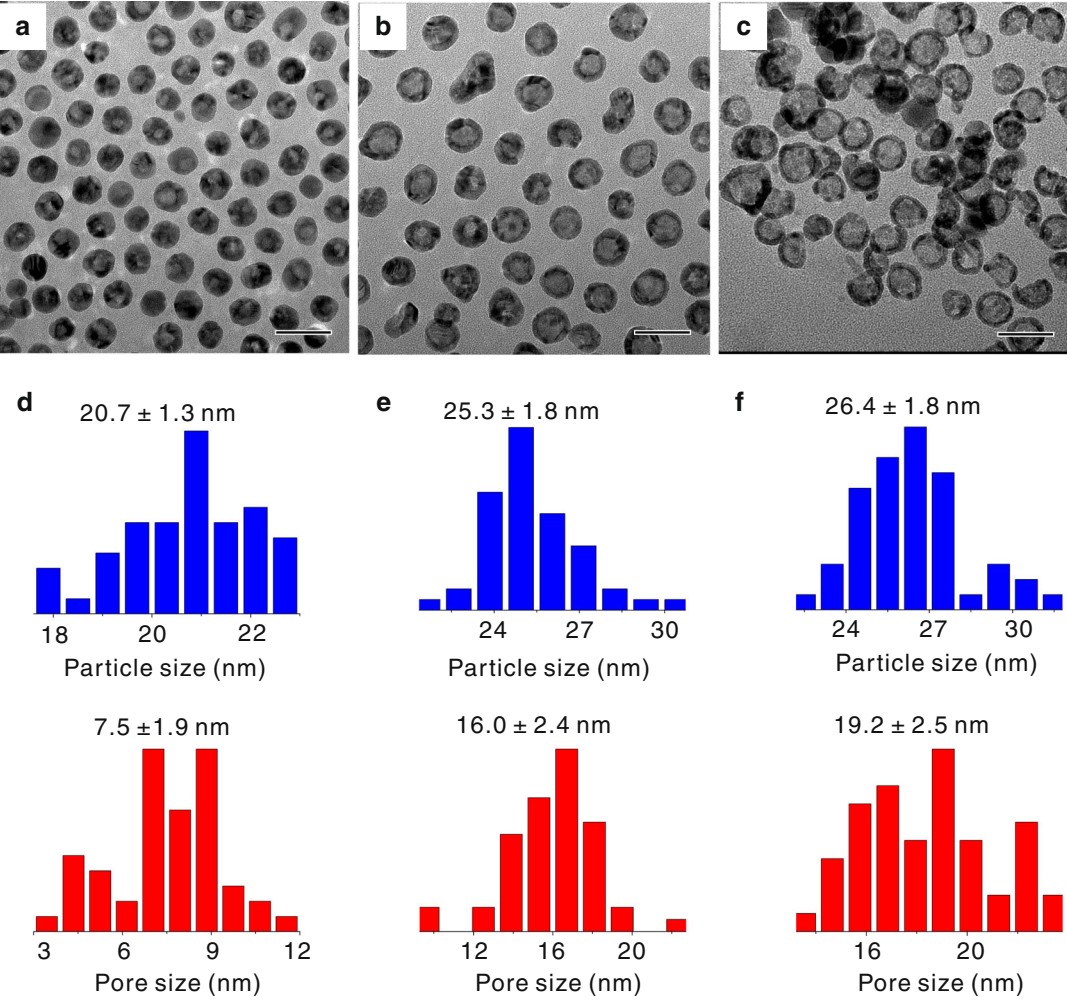

**Fig. 5** Size variation of the hollow nanocrystals obtained after repeated Kirkendall cavitation processes. TEM images and statistical analyses of particle size and pore size distribution for **a**, **d** H-Pd-1, **b**, **e** H-Pd-2, and **c**, **f** H-Pd-3. Scale bars in **a–c** are 50 nm

transformed into hollow nanocrystals with almost no loss of Pd atoms. In principle, the process could be further repeated to obtain hollow nanocrystals with even thinner shells; however, in practice, this would become challenging as thinner shells have higher tendency to rupture.

**Catalytic property of hollow Pd nanocrystals**. One of the advantageous applications of hollow nanocrystals with thin shells is in catalysis as they provide more accessible active sites and thus improve atomic utilization efficiency[20, 21]. Here, we use a formic acid oxidation (FAO) reaction as a model reaction to evaluate the catalytic performance of the hollow Pd nanocrystals generated after repeated Kirkendall cavitation processes, with a commercial Pd/C (10 wt%) sample for benchmarking purpose. High-angle annular dark-field scanning TEM (HAADF-STEM) images of H-Pd-2 and H-Pd-3 showed that there were some pores on the shells, making hollow Pd nanocrystals permeable during catalytic reactions (Supplementary Fig. 14). Therefore, both the inner and outer surfaces of the hollow Pd nanocrystals can take part in the catalytic reactions, thus enhancing the catalytic activity of hollow Pd nanocrystals. In addition to the enlargement of the surface area, surface roughness and defects can also play an important role in enhancing the catalytic activity of surface atoms[46–48]. Supplementary Figure 15 shows the HRTEM images of H-Pd-2 and H-Pd-3. Clearly, the surface roughness and the number of

defects increased after the transformation. As demonstrated in Fig. 6a, b, H-Pd-1 exhibited a recessive mass activity relative to Pd/C, while H-Pd-2 and H-Pd-3 outperformed Pd/C regarding both specific activities and mass activities despite their relatively larger particle sizes. Specifically, H-Pd-3 showed a mass activity of 1800 mA mg$^{-1}$ at 0.22 V vs. Ag/AgCl electrode, almost four times the value of Pd/C. Meanwhile, the specific activity demonstrated by H-Pd-3 (1800 µA cm$^{-2}$) was also over three times that of Pd/C (Supplementary Fig. 16). The increased number of accessible atoms due to continuous thinning of the shells is believed to be responsible for the improved catalytic performance. When the shell is relatively thick, the inner Pd atoms are less accessible. Thus, the size effect would give the Pd/C a better catalytic performance in terms of mass activity due to a larger surface to volume ratio. When the nanocrystals are inflated to ones with extremely thin shells, the contribution from inner atoms can produce more active reaction sites, thus effectively promoting the catalytic performance. The electrochemically active surface areas (ECSAs) calculated from the carbon monoxide (CO) stripping cyclic voltammetry curves indicated an increasing order of H-Pd-1 < Pd/C < H-Pd-2 < H-Pd-3, further verifying the inference above. Finally, we tested the durability of the catalysts. As shown in Fig. 6d, the activity of Pd/C dropped by 75% after 1000 cycles of the FAO reaction, while over 60% of the initial activities were sustained for all hollow Pd catalysts after 1000 cycles. Postmortem examination of H-Pd-3 indicated that the hollow

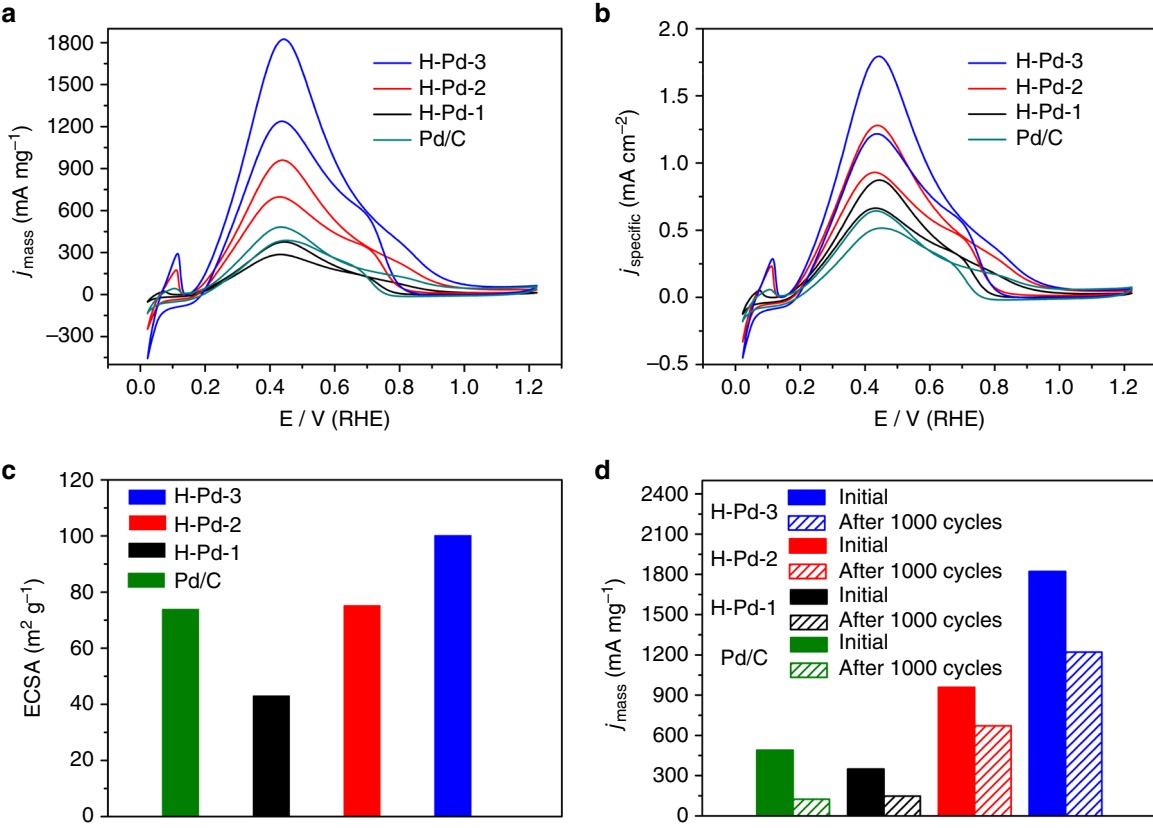

**Fig. 6** Electrochemical catalytic performance of hollow Pd nanoparticles and commercial Pd/C toward formic acid oxidation. **a** Mass and **b** specific activities of the catalysts. **c** Specific electrochemically active surface areas of the catalysts. **d** Mass activities of the catalysts before and after the durability test

structures were well preserved after 1000 cycles, suggesting a high stability, which could be attributed to the reduced tendency of sintering of the hollow nanocrystals due to their relatively larger overall dimensions than those in the commercial sample. This excellent stability is consistent with previous reports on hollow/frame structured metallic nanocrystals[49, 50]. Collectively, these results demonstrate the great potential of this repeated Kirkendall cavitation process in engineering the structures of nanoscale metallic catalysts for enhanced catalytic performance.

In summary, we have demonstrated a repeatable nanoscale Kirkendall cavitation process for the transformation of solid Pd nanocrystals into hollow Pd nanocrystals. Through repeated processes of insertion and extraction of P, the nanoscale Kirkendall effect drives the "inflation" of the Pd hollow nanocrystals by progressively increasing their outer diameter and reducing the shell thickness. The resulting thin shells exhibit increased catalytic activity and durability toward FAO reaction, demonstrating mass activity and specific activity four and three times higher than those of commercial Pd/C catalysts, so as to satisfy the requirements for efficient electrocatalytic applications. The Kirkendall cavitation process has been confirmed theoretically by atomistic simulation studies. We believe the current study could significantly broaden the application of the Kirkendall effect in the rational design of nanoscale structures, produce practically important catalysts with improved activity and durability, and thus impact broad areas such as fine chemical production, environmental protection, and energy conversion.

## Methods

**Chemicals**. All chemicals involved in the present work, including sodium tetra-chloropalladate ($Na_2PdCl_4$, 98%), poly (vinyl pyrrolidone) (PVP, molecular

weight ≈ 55,000), ascorbic acid (99%), tri-$n$-octylphosphine (TOP, 90%), potassium bromide (KBr, 99%), oleylamine (OAm, 80–90%), acetic acid ($CH_3COOH$, 99.7%), formic acid (HCOOH, 98%), and perchloric acid ($HClO_4$, 70%), were purchased from Sigma-Aldrich and used without any further modification.

**Synthesis of Pd nanocubes**. In a typical synthesis, 11 mL of an aqueous solution containing poly(vinyl pyrrolidone) (PVP, molecular weight≈55,000, 105 mg, Aldrich),
L-ascorbic acid (60 mg, Aldrich), KBr (600 mg, Fisher), and sodium tetra-chloropalladate ($Na_2PdCl_4$, 57 mg, Aldrich) was heated at 80 °C in air under magnetic stirring for 3 h. The nanocubes were collected by centrifugation, washed three times with deionized water, and eventually re-dispersed in 11 mL of deionized water at a concentration of 1.68 mg mL$^{-1}$.

**Synthesis of hollow Pd nanocrystals**. For the synthesis of Pd hollow nanocrys-tals, 22 mL of OAm, 2 mL of TOP, and 2 mL of an aqueous suspension (1.68 mg mL$^{-1}$) of the 18-nm Pd cubes were mixed in a 50-mL three-neck round-bottom flask, and heated under magnetic stirring at 250 °C for 260 min under the flow of nitrogen gas before it was cooled to room temperature. As the reaction took place at 250 °C, water in an aqueous suspension of Pd nanocubes evaporated from the reaction system, and the nanocubes were found to be well dispersed in OAm. Then the solution was divided into four parts, in each of which the nanoparticles were collected by centrifugation and washed twice with ethanol. One part of the sample was re-dispersed in 6 mL of OAm under sonication, transferred to a 50-mL three-neck round-bottom flask, and heated at 250 °C for 80 min under the air. For the control of the reaction atmosphere, different volumes of $O_2$ and $N_2$ gases were firstly mixed in a 2-L gas sampling bag. Then, the reaction vial was evacuated and connected to the gas sampling bag to control the reaction atmosphere. When cooled to room temperature, the product, denoted as H-Pd-1, was collected by centrifugation and washed twice with ethanol. The phosphorization and depho-sphorization reactions could be repeated by following the above procedure, except by using H-Pd-1 or H-Pd-2 as the starting materials.

**Evaluation of electrochemical catalytic activity**. Hollow Pd hollow nanocrystal (H-Pd-1, H-Pd-2, and H-Pd-3) were deposited onto Ketjen Carbon (C) (EC300J) by magnetically stirring the mixture of nanoparticle and 4.5 mg carbon in 5 mL acetic acid and for 24 h, washing the solid sample twice with hexane, and drying the

powder at 180 °C for 12 h. Then, the carbon-supported catalysts were dispersed in a mixture containing 2.25 mL ethanol and 0.25 mL 5% Nafion. Then, 20 µL of the suspension was added to a glassy carbon rotating disk electrode and dried in air. For benchmarking purpose, 2 mg commercial Pd on carbon (10 wt%) was dispersed in 1 mL solution composed of 0.9 mL ethanol and 0.1 mL 5% Nafion, after which 20 µL of the suspension was deposited on a pre-cleaned glassy carbon rotating disk electrode.

The CO stripping method was conducted to determine the ECSAs of all catalysts, with an Ag/AgCl electrode and a platinum foil as the reference and counter electrodes, respectively. The working electrode was immersed in a 0.5 M $H_2SO_4$ solution with CO gas (99.99%) bubbling for 20 min and then quickly moved to another fresh solution, after which the CO stripping voltammetry was recorded at a sweep rate of 50 mV s$^{-1}$.

In the electrooxidation of formic acid, cyclic voltammograms were recorded at a sweep rate of 50 mV s$^{-1}$ in a 150 mL solution containing 0.5 M $HClO_4$ and 0.5 M formic acid. Two cycles of potential sweep were carried out between −0.2 V and 1.2 V at a sweep rate of 250 mV s$^{-1}$ before the cyclic voltammetry measurements.

**Sample characterizations**. TEM images were recorded on a Hitachi HT-7700 microscope equipped with a tungsten filament, operating at 100 kV. HRTEM imaging, HAADF-STEM imaging, and EDS elemental mapping were performed on a JEM-2100F (JEOL) equipped with a built-in EDS. The percentages of Pd and Pt in the samples were determined using ICP-MS (Agilent 7500ce ICP-MS). Powder XRD patterns were recorded using a diffractometer (X-ray Diffractometer Smar-tLab(3), Rigaku) operated at 3 kW.

**Data availability**. The data that support the findings of this study are available within the article (and its Supplementary Information files) and from the corresponding authors on reasonable request.

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

## Acknowledgements

M.J. thanks the National Natural Science Foundation of China (NSFC, nos. 21471123 and 21403160) and the "start-up fund" provided by Xi'an Jiaotong University. Y.Y. thanks the financial support from the US National Science Foundation (CHE-1308587). X.Y. and Z.W. thanks the financial support from the Guangxi Science Foundation (grant no. 2013GXNSFFA019001), the Guangxi Key Laboratory Foundation (grant no. 15-140-54), and the Scientific Research Foundation of Guangxi University (grant no. XTZ160532).

## Author contributions

Y.Y. and M.J. conceived and designed the experiments. T.H. and W.W. performed the catalyst preparation, catalytic testing, characterization, and wrote part of the paper. X.Y. and Z.W. carried out atomistic simulation for this work. Z.C. and Q.K. characterized the PdP$_x$ nanospheres and analyzed the data. Z.S. contributed to the microscopy analysis. Y.Y. and M.J. wrote and revised the paper. All authors discussed the results and commented on the manuscript.

## Additional information

**Competing interests:** The authors declare no competing financial interests.

