## [Peer Review File · Nature Communications]

Reviewers' Comments:

Reviewer #1:

Remarks to the Author:

The manuscript contributed by He et al. reports a very clever way to synthesize hollow metal nanoshells with tunable wall thicknesses and void sizes through the repeated insertion/extraction of an extra element to alloy with the metal. Selective oxidation of the added element and nanoscale Kirkendall process ensures the success, which cannot be imagined in the past. There is no doubt that this clever approach represents a most promising advance in recent years towards the controlled synthesis of hollow metal nanoparticles. It is so amazing to see the unique "inflating" process works very well at the nanometer scale and it is applied to synthesize useful materials for efficient catalysis.

The synthesized hollow Pd nanoparticles with varying thicknesses and void sizes exhibit significantly improved performance (in terms of activity and stability) towards oxidation of formic acid. The enhanced performance is impressive. There is no doubt that this manuscript can bring a broad impact on all fields related to functional nanomaterials.

Clarifying the following minor questions can further improve the clarity of the manuscript. What parameters make the major contributions to the improved performance? Of course, inflating the Pd nanoparticles can increase exposed surface area. Is the activity directly proportional to the outer surface area of the hollow Pd nanoparticles? Otherwise, do the inner surfaces of the hollow Pd nanoparticles also contribute to the improved activity? If it is possible for the inner void surfaces to serve as active catalytic sites, the Pd shells should be permeable. It is suggested to characterize the crystallinity of the Pd shells to justify the permeability of the Pd shells.

Overall, the reviewer strongly supports to publish this amazing work.

Reviewer #2:

Remarks to the Author:

The authors demonstrate a repeatable nanoscale Kirkendall cavitation process for generating hollow Pd nanocrystals via inserting and extracting P. By repeating this nanoscale Kirkendall effect, the outer diameter of the nanoparticles gradually increases while shell thickness progressively decreases. This study could significantly broaden the application of the Kirkendall effect in the rational design of hollow nanostructures. I suggest the publication of this paper after major revisions:

- 1) The authors claimed that after P extraction with oxygen, the P:Pd ratio decreased to 1:8.8 and the remaining P was from the TOP molecules. The authors need to confirm this point of view with more experiments. If it was TOP, after ethanol cleaning, the amount should significantly decrease. The authors should use FT-IR to detect whether the P was in a molecular state or not.
- 2) The authors should provide the HAADF-STEM EDX mapping results of the PdP₂ intermediates to demonstrate whether the P was within the entire particle or just concentrated on the surface of the particles.
- 3) The authors begin with Pd nanocubes which transform to spherical particles during the first conversion. The authors need to discuss this transformation. Is the selection of Pd nanocubes (or single-crystalline particles) to being this process important to the synthetic outcome?
- 4) The change in particle size and size dispersity is important to the utility of this synthetic approach. The only quantitative information on the changes in particle size is buried in Figure S7 and not discussed in the main text. This information needs to be moved into the main text and applied to more samples to quantitatively show the effect of O₂ content in the synthesis and also for the enlargement of cavities.
- 5) My biggest concern with this manuscript is the experimental reporting. Mainly, the oxygen concentration is critical to the cavitation process, but the oxygen concentration is not quantified

and the experimental section does not provide sufficient information to reproduce these results. There is especially an absence of information about how the authors changed the atmosphere experimentally and quantified the oxygen concentration.

Reviewer #3:

Remarks to the Author:

Dear Editor,

HE et al presents a beautiful work where the plasticity of inorganic nanocrystals is shown in a reversible and incremental process of alloying and dealloying cycles controlling competitive diffusion rates altering the environmental temperature and oxygen concentration.

The inhomogeneity in the final NCs could be related to inhomogeneity in PdP2 NCs ... because of that, and given the actual characterization power I would strongly suggest authors to include EELS/EDAX elemental mapping to observe the homogeneity of the atomic distribution. Also, it would be interesting the role of P2O5 which when formed reacts with water making ($P_4O_{10} + 6H_2O = 4H_3P_2O_4$) and consequently altering the acidity of the solution what has strong effects on oxidation processes. Authors should comment on that.

I am not convinced about the ultra-thin shells... regarding the data they are as thin as any other hollow NP, between 3 and 5 nm... this are not ultra-thin but standard in nanoshells/hollow nanocrystals.

In line 176 they claim that as the NP expand and you add no more Pd forcedly the shell thickness will thin as the diameter increases... this is so evident... I do not see the point of this statement... off course that as you make the NP bigger with the same mass, shell has to be thinner... this is why it is called "balloon"... they do so... again, in line 202 appears that authors discover that with the same amount of Pd, making thinner shells in larger hollow NPs will increase surface area... this is also very evident. Regarding this point, and with the hi resolution TEM work, one could compare atomic surface roughness/structure to correlate it with the increased catalytic activity.

ALSO IT WOULD BE VERY IMPORTANT that they also plot catalytic activity by surface area (and not by mass) so in this way one can conclude if the enhanced activity is because they have the same surface for less grams or if the surface of the hollow inflated NCs are more reactive than the solid equivalent ones... in this regard, atomic sub-coordination leading to increased catalytic activity may also result in structural weakness in such a way that after few runs the wonderful catalyst becomes poisoned or inactivated... this was recently described in Pt hollow NCs [<http://pubs.rsc.org/en/content/articlelanding/2015/ta/c5ta07504a> #!divAbstract].

In line 190 they say that catalytic activity depends on surface structure of atoms and in "size"... I am not sure what do authors refer here to... if it is about electromagnetic quantum confinement effects it happens for very small sizes in metals, besides the effect of increased curvature radii when decreasing size is translated into less coordination of atoms at the surface and higher catalytic activity... but this is again a surface organization atoms... later on they talk about habits and facets... and this is more related to shape than size... authors should clarify this point.

After addressing this points the paper would be ready to make a nice contribution to Nat Comm

Referee #1:

“1) Clarifying the following minor questions can further improve the clarity of the manuscript. What parameters make the major contributions to the improved performance? Of course, inflating the Pd nanoparticles can increase exposed surface area. Is the activity directly proportional to the outer surface area of the hollow Pd nanoparticles? Otherwise, do the inner surfaces of the hollow Pd nanoparticles also contribute to the improved activity? If it is possible for the inner void surfaces to serve as active catalytic sites, the Pd shells should be permeable. It is suggested to characterize the crystallinity of the Pd shells to justify the permeability of the Pd shells.”

We thank this reviewer for his/her constructive question. As suggested, we further carried out HAADF-STEM and HRTEM analyses to characterize the crystallinity of the Pd shells, and the results were added in the Supporting Information as Figures S12 and S13. HAADF-STEM images of H-Pd-2 and H-Pd-3 showed that there were some pores on the shells, making hollow Pd nanocrystals permeable during catalytic reactions (Figure S12). Therefore, both the inner and outer surfaces of hollow Pd nanoparticles can take part in the catalytic reactions, thus significantly enhance the catalytic activity of hollow Pd nanocrystals. In addition to the enlargement of the surface area, surface roughness and defects can also play important role in enhancing the catalytic activity of surface atoms. Figure S13 shows the HRTEM images of H-Pd-2 and H-Pd-3. Clearly, the surface roughness and number of defects increased after the transformation of nanocubes to hollow nanocrystals. In order to clarify these issues, some discussions and references have been added in the revised manuscript (Please see the highlighted sentences in the first paragraph in Page 11, refs. 46-48, Figures S12 and S13).

Referee #2:

“1) The authors claimed that after P extraction with oxygen, the P:Pd ratio decreased to 1:8.8 and the remaining P was from the TOP molecules. The authors need to confirm this point of view with more experiments. If it was TOP, after ethanol cleaning, the amount should significantly decrease. The authors should use FT-IR to detect whether the P was in a molecular state or not.”

Thanks for this reviewer's good question. After washing with ethanol twice, the ratio of P:Pd was found to further decrease to 1:11.8. We also used FT-IR to characterize hollow Pd nanocrystals, and the result was added in the Supporting Information as Figure S5. As shown in this new result, all the peaks can be indexed to the characteristic absorbance of TOP, implying that P might still exist in the form of TOP molecules which were capped on the surface of hollow Pd nanocrystals. The relevant discussions have been added to the revised manuscript. Please see the highlighted sentences in the first paragraph in Page 6.

“2) The authors should provide the HAADF-STEM EDX mapping results of the PdP2 intermediates to demonstrate whether the P was within the entire particle or just concentrated on the surface of the particles.”

As suggested, HAADF-STEM EDX were used to characterize elemental distribution within the PdP2 intermediates, and the results were added in the Supporting Information as Figures S2 and S3. The EDX elemental mapping images and line-scanning profiles of the PdP2 intermediates demonstrated

that both Pd and P distributed homogeneously inside within the particles. In the revised manuscript, some sentences and Figures S2 and S3 were added. Please see the highlighted sentences in Page 5, and Figures S2 and S3.

“3) The authors begin with Pd nanocubes which transform to spherical particles during the first conversion. The authors need to discuss this transformation. Is the selection of Pd nanocubes (or single-crystalline particles) to being this process important to the synthetic outcome?”

Thanks for this good question. The selection of single-crystalline particles is not crucial for the inflating process. We chose nanocubes as the starting materials simply because we have a good recipe for producing them in a very uniform manner. Twinned nanoparticles, such as decahedrons, can also be introduced as starting materials to generate hollow nanocrystals. We have carried out additional reactions using Pd decahedrons as the starting materials, and produced hollow nanocrystals as shown in Figure S7.

The shape transformation from nanocubes to nanospheres can be attributed to the significant lattice disruption when P atoms were inserted into the original Pd nanocubes.

The relevant discussion has been added to the revised manuscript (see the highlighted sentences in the first paragraph of Page 5, and the first paragraph of Page 7).

“4) The change in particle size and size dispersity is important to the utility of this synthetic approach. The only quantitative information on the changes in particle size is buried in Figure S7 and not discussed in the main text. This information needs to be moved into the main text and applied to more samples to quantitatively show the effect of O₂ content in the synthesis and also for the enlargement of cavities.”

We thank this reviewer for this kind suggestion. Accordingly, we have moved Figure S7 to the main text as Figure 5. The related quantitative analyses of the samples prepared with different O₂ content were also added in the revised manuscript, see Figure S9. To better illustrate these issues, more discussion was also added to the main text. Please see the highlighted sentences in the last paragraph in Page 8, and first paragraph of Page 10.

“5) My biggest concern with this manuscript is the experimental reporting. Mainly, the oxygen concentration is critical to the cavitation process, but the oxygen concentration is not quantified and the experimental section does not provide sufficient information to reproduce these results. There is especially an absence of information about how the authors changed the atmosphere experimentally and quantified the oxygen concentration.”

We are sorry for the unclear description of our experiments. During revision, we have modified the experimental section. The detailed information, including the control of the reaction atmosphere, has been added in the revised manuscript. Please see the highlighted sentences in the first paragraph of Page 14.

Referee #3:

“1) The inhomogeneity in the final NCs could be related to inhomogeneity in PdP2 NCs ... because of that, and given the actual characterization power I would strongly suggest authors to include EELS/EDAX elemental mapping to observe the homogeneity of the atomic distribution.”

We thank the reviewer for this kind suggestion. Accordingly, we carried out HAADF-STEM EDX elemental analyses for the PdP2 intermediates, and included the results in the Supporting Information as Figures S2 and S3. The EDX elemental mapping images and line-scanning profiles of the PdP2 intermediates demonstrated that both Pd and P distributed homogeneously within the particles. In order to clarify the elemental distribution, some sentences, as well as Figures S2 and S3 were added in the revised manuscript. Please see the highlighted sentences in the first paragraph of Page 5.

“2) Also, it would be interesting the role of P2O5 which when formed reacts with water making ($P_4O_{10} + 6H_2O = 4 H_3P_2O_4$) and consequently altering the acidity of the solution what has strong effects on oxidation processes. Authors should comment on that.”

The reaction was carried out in the absence of water as both the phosphorization and oxidation reactions took place at 250 °C with OAm as the solvent. Upon the formation of P2O5, OAm will react with it. As OAm is the solvent and it is in large excess, the produced P2O5 is not expected to cause apparent change to the acidity of the solution, and therefore should have minimal effect on the oxidation processes. We have added this comment in the revised manuscript. Please see the highlighted sentences in the first paragraph of Page 6.

“3) I am not convinced about the ultra-thin shells... regarding the data they are as thin as any other hollow NP, between 3 and 5 nm... this are not ultra-thin but standard in nanoshells/hollow nanocrystals.”

We thank the reviewer for pointing out this issue. Accordingly, we have changed “ultra-thin” to “thin” in the revised manuscript.

“4) In line 176 they claim that as the NP expand and you add no more Pd forcedly the shell thickness will thin as the diameter increases... this is so evident... I do not see the point of this statement... off course that as you make the NP bigger with the same mass, shell has to be thinner... this is why it is called “ballon”... they do so... again, in line 202 appears that authors discover that with the same amount of Pd, making thinner shells in larger hollow NPs will increase surface area... this is also very evident. Regarding this point, and with the hi resolution TEM work, one could compare atomic surface roughness/structure to correlate it with the increased catalytic activity.”

Thanks for this good suggestion. We have added these statements in the revised manuscript. Indeed, the shell thickness decreases as the diameter of hollow Pd nanocrystals increases, since no additional Pd source were introduced into the reaction system (Figure 5). In addition to the enlargement of the surface area, surface roughness and defects can also play important roles in enhancing the catalytic activity of surface atoms. Figure S13 shows the HRTEM images of H-Pd-2 and H-Pd-3. Clearly, the surface roughness and number of defects increased after the transformation of nanocubes to hollow nanocrystals. We have added some relevant discussions in the revised manuscript. Please see the highlighted sentences in the first paragraph of Page 11. In addition, some related references were also added, see Refs. 46-48.

“5) ALSO IT WOULD BE VERY IMPORTANT that they also plot catalytic activity by surface area (and not by mass) so in this way one can conclude if the enhanced activity is because they have the same surface for less grams or if the surface of the hollow inflated NCs are more reactive than the solid equivalent ones... in this regard, atomic sub-coordination leading to increased catalytic activity may also result in structural weakness in such a way that after few runs the wonderful catalyst becomes poisoned or inactivated... this was recently described in Pt hollow NCs [http://pubs.rsc.org/en/content/articlelanding/2015/ta/c5ta07504a#!divAbstract].”

Thanks for pointing this out. In fact, the specific activity (catalytic activity by surface area) in electrochemical catalysis of FAO was provided in Figure 5b of the original manuscript (now Figure 6b in the revised version). The results clearly show that the enhanced activity of the hollow inflated nanocrystals is not only due to the enlarged surface area. To make a better comparison, we have plotted the specific activities (catalytic activity/surface area) as a histogram for all the samples. Please see the Figure S14 in the Supporting Information. As stated previously, we believe that the enhancement of catalytic activity of the inflated sample is not only due to enlargement of the surface area, but also higher surface roughness and more defects.

We thank the reviewer for pointing out the possibility of poisoning/inactivation of the catalyst due to structural weakness. In our original manuscript, we include the results on the stability of the catalysts. As can be seen in Figure 6d, after running for 1000 cycles, the activity of hollow Pd nanoparticles decreased about 40%. Although some inactivation was observed, the catalyst could still be considered to be quite robust as the activity of Pd/C decreased 75% after similar conditions. This comparison indicates the excellent stability of hollow nanoparticles compared with commercial Pd/C catalysts. This result is also consistent with previous reports on hollow/frame structures. We have added relevant discussions to the revised manuscript. Please see the highlighted sentences in the first paragraph of Page 12, and refs. 49, 50.

“6) In line 190 they say that catalytic activity depends on surface structure of atoms and in “size”... I am not sure what do authors refer here to... if it is about electromagnetic quantum confinement effects it happens for very small sizes in metals, besides the effect of increased curvature radii when decreasing size is translated into less coordination of atoms at the surface and higher catalytic activity... but this is again a surface organization atoms... later on they talk about habits and facets... and this is more related to shape than size... authors should clarify this point.”

Thanks for the question. By “size” we were referring to the general understanding that nanocrystals with smaller sizes have larger surface to volume ratio, which can provide more active sites for catalytic reactions. Therefore, “size” here is not about the electromagnetic quantum confinement effects. Along with the decrease of the shell thickness, surface roughness and number of defects increase, thereby resulting in higher catalytic activities of surface atoms. In order to clarify the meaning of “size”, some sentences have been added in the revised manuscript. Please see the highlighted sentences in the first paragraph of Page 10 and the first paragraph of Page 11.

Reviewers' Comments:

Reviewer #1:

Remarks to the Author:

The revised manuscript with additional supporting information clearly addresses the questions raised in the review process. It is now suitable for being published.

Reviewer #2:

Remarks to the Author:

I am satisfied with the additional experiments and revisions to the manuscript.

Reviewer #3:

Remarks to the Author:

The work has been significantly improved and it is ready for publication

Reviewers' comments:

“REVIEWERS' COMMENTS:

Reviewer #1 (Remarks to the Author):

“The revised manuscript with additional supporting information clearly addresses the questions raised in the review process. It is now suitable for being published.”

Reviewer #2 (Remarks to the Author):

I am satisfied with the additional experiments and revisions to the manuscript.

Reviewer #3 (Remarks to the Author):

The work has been significantly improved and it is ready for publication”

Again, we thank these reviewers for their great efforts in helping to improve the quality of this work.